# Computation of X-ray and Neutron Scattering Patterns to Benchmark Atomistic Simulations against Experiments

**DOI:** 10.3390/ijms25031547

**Published:** 2024-01-26

**Authors:** Arnab Majumdar, Martin Müller, Sebastian Busch

**Affiliations:** 1German Engineering Materials Science Centre (GEMS) at Heinz Maier-Leibnitz Zentrum (MLZ), Helmholtz-Zentrum Hereon GmbH, Lichtenbergstr. 1, 85748 Garching, Germany; arnab.majumdar@hereon.de (A.M.); martin.mueller@hereon.de (M.M.); 2Institute of Materials Physics, Helmholtz-Zentrum Hereon GmbH, Max-Planck-Str. 1, 21502 Geesthacht, Germany; 3Institut für Experimentelle und Angewandte Physik (IEAP), Christian-Albrechts-Universität zu Kiel, Leibnizstr. 19, 24098 Kiel, Germany

**Keywords:** neutron scattering, X-ray scattering, quasielastic neutron scattering, wide-angle diffraction, small-angle scattering, molecular dynamics simulation, finite-size effect, Sassena

## Abstract

Molecular Dynamics simulations study material structure and dynamics at the atomic level. X-ray and neutron scattering experiments probe exactly the same time- and length scales as the simulations. In order to benchmark simulations against measured scattering data, a program is required that computes scattering patterns from simulations with good single-core performance and support for parallelization. In this work, the existing program Sassena is used as a potent solution to this requirement for a range of scattering methods, covering pico- to nanosecond dynamics, as well as the structure from some Ångströms to hundreds of nanometers. In the case of nanometer-level structures, the finite size of the simulation box, which is referred to as the *finite size effect*, has to be factored into the computations for which a method is described and implemented into Sassena. Additionally, the single-core and parallelization performance of Sassena is investigated, and several improvements are introduced.

## 1. Introduction

The purpose of this work is to show that neutron and X-ray scattering experiments are potent tools for the validation of atomistic simulations. The fundamental idea behind this is that scattering techniques probe the atomic structure and dynamics on relevant scales, both in time and in space [1,2]—the scattering curves can be directly computed from the time-dependent positions of the nuclei in the simulation. This contribution focuses on a fast and reliable method of this computation, opening up the opportunity for the validation of force fields or even fitting the force field parameters.

Two techniques lend themselves particularly well to be used for atomistic simulations: Monte Carlo (MC) [3,4] and Molecular Dynamics (MD) [5,6]. While MC simulations can only capture the structure of a sample, MD simulations give access to structure and dynamics [7]. Both techniques are used to explore representative states of the system, which are often determined by interatomic potentials (the force field). The closeness of the simulation to reality has been gauged in a range of works by comparing the scattering curves computed from the simulations to experimental scattering data (e.g., [8,9,10,11,12,13,14]). If the solvent is not modeled explicitly in a solute/solvent-type system, the calculation of the corresponding scattering curves has to pay special attention to the non-uniform density of the solvent close to the solute [15,16,17].

Attempts were also made to optimize this agreement. Among several attempts, particularly notable ones are the Monte Carlo-based Reverse Monte Carlo (RMC) [6,18] and Empirical Potential Structure Refinement (EPSR) [19] for amorphous materials, as well as RMCProfile [20] for polycrystalline materials. Approaches using Molecular Dynamics simulations have been applied to nanometer-sized objects [21,22], liquids in porous media [23], and polymers [24], to name a few.

Such a fit of the simulation to the scattering data necessitates the repeated computation of the MD trajectory and scattering patterns, namely at each iteration of the optimization algorithm with updated potentials. Speed and energy efficiency are therefore essential for the involved programs, also in view of sustainable operation [25]. In recent years, hardware development saw a leveling off of the processor clock speed with more and more cores embedded in one processor [26]. In order to use this hardware at full capacity, programs need to support parallelization [27].

For the MD simulation itself, programs like LAMMPS [28] or GROMACS [29] ensure this and make it possible to simulate billions of atoms [30] and times up to the range of microseconds [31,32] with a supercomputer. With this increase in the number of atoms and time steps, it is of prime importance to optimize the software to convert trajectories to scattering patterns as much as possible. The most pertinent software options are currently nMoldyn [33,34], MDANSE [35,36], LiquidLib [37,38], and Sassena [39,40,41,42]. The requirement for speed restricts the choice to the latter two [43], and if not only amorphous systems are to be studied, only Sassena remains. In contrast to Sassena, LiquidLib and MDANSE can also calculate other quantities from the simulations, but there are also many other software options like VMD [44,45], MDAnalysis [46,47], TRAVIS [48,49], or viamd [50] for this purpose. The focus of this work is therefore the calculation of neutron and X-ray scattering patterns using and enhancing Sassena.

Even Sassena would take an estimated 25 years to compute the diffractogram of a billion atoms simulation on one core. Therefore, a rather technical enhancement described in this contribution concerns the careful use of parallelization on different levels of the program, which results in a significant speed improvement.

A more fundamental issue is related to the finite size of the simulation box: In the real world experiment, the “box” is virtually infinitely large, and its shape will therefore not cause any measurable scattering signal. Due to the aforementioned computational limitations, the actual simulation boxes are preferably as small as need be, which leads to a very pronounced spurious signal in the small-angle region of the computed scattering curve due to a *finite size effect* [51,52,53,54]. It complicates the comparison with the experimental data. This spurious signal was removed in an innovative way: while the existing solution for this task, which will be referred to as *r-clean*, distorts the wide-angle region of diffraction patterns after removing the spurious signal from the small-angle region, our new method *Q-clean* does not introduce any distortions.

In this work, we will demonstrate the capabilities of Sassena [42] by computing the relevant quantities for the investigation of dynamics at the atomic level, and for the investigation of structure at the atomic and nanometer levels with several examples. Fast calculations are ensured by optimized single-core and multi-core performance.

## 2. Results

We aim to use Sassena for the computation of scattering curves from all kinds of classical MD simulations and benchmark them against experimental scattering data. Some representative examples of benchmarking the molecular structure and dynamics are shown in this section, namely (a) pure water, exhibiting dynamics at the atomic level; (b) mixtures of normal and heavy water, exhibiting structure at the atomic level but homogeneity at the nanoscopic scale; (c) a thought experiment of a spherical solute in solvent as a toy example of a well-defined structure at the nanometer level; and (d) the protein lysozyme in water as a real-world case of a nanometer-sized object.

In order to ensure the fast computation speed of Sassena, the impact of the most important parameters on the computation times is investigated, and several optimizations are included in Sassena to boost its single-core and parallelization (or multi-core) performance.

### 2.1. Atomistic Dynamics as Seen by the Incoherent Intermediate Scattering Function

The dynamics, i.e., the motions in a sample, can be determined by measuring the transfer of kinetic energy between sample atoms and the probe in a *spectroscopic* experiment: the scattered X-ray/neutron intensity is recorded as a function of the momentum transfer or scattering vector $\vec{Q}$, after orientational averaging, simply its absolute value *Q*, and the energy transfer ℏω, or equivalently its Fourier-partner correlation time τ. Without being a substitute for the Fourier transform needed to switch between real and reciprocal space, a useful rule of thumb is that the scattering data at a given *Q* are mainly sensitive to distances on the order of r∼2π/Q. Similarly, the time scale in picoseconds of processes causing an energy transfer ℏω in meV can be estimated to be τ∼0.658·2π/(ℏω), where the factor 0.658 is due to the unit conversions. The pico- to nanosecond time range relevant to classical atomistic MD simulations tends to be a forte of neutron scattering. In order to compare the simulations to scattering data, intermediate scattering functions I(Q,τ) were calculated from the simulated trajectories as shown in Figure 1.

In the naming convention used in this contribution, X-rays are only scattered *coherently*, while neutrons can also be scattered *incoherently*—terms that will be connected to formulas below. Atoms of different elements—in the case of neutrons, even different isotopes—scatter differently, which is quantified by a scattering strength factor f(Q). For neutrons, $f^{\mathrm{coh},\mathrm{neutron}}$ and $f^{\mathrm{inc},\mathrm{neutron}}$ are *Q*-independent constants $b^{\mathrm{coh}}$ and $b^{\mathrm{inc}}$, named the *scattering length*. In the case of X-rays, the *atomic form factor*
$f^{\mathrm{coh},\mathrm{xray}}\left(Q\right)$ is usually approximated by a sum of Gaussians. The values of the neutron scattering lengths are tabulated [56] as are the coefficients of the Gaussians that make up the X-ray atomic form factor [57].

While Sassena can calculate the coherent and incoherent intermediate scattering function as discussed in Appendix A, this section will concentrate on the incoherent intermediate scattering function because, in the example of water, the hydrogen causes so much incoherent scattering that the whole measured signal can be approximated as pure incoherent scattering [58]. Since incoherent scattering does not require the preservation of the structure, the output trajectory from the MD simulation is saved in the *unwrapped* format, where atoms are allowed to move to their periodic images. This prevents any sudden jumps of atoms due to periodic boundary conditions. Details about the simulation of the system are described in Appendix B. The incoherent intermediate scattering function $I^{\mathrm{inc}}\left(Q,\tau \right)$, calculated from the output trajectory, is
(1)$$I^{\mathrm{inc}}\left(Q,\tau \right)={\langle \sum_{j=1}^{N}{\left|b_{j}^{\mathrm{inc}}\right|}^{2}\exp\left[i\vec{Q}\cdot \left({\vec{R}}_{j}\left(t+\tau \right)-{\vec{R}}_{j}\left(t\right)\right)\right]\rangle }_{t,\Omega }\;,$$
where τ is the correlation time, $b_{n}^{\mathrm{inc}}$ the incoherent neutron scattering length of the nth atom, ${\vec{R}}_{n}\left(t\right)$ is the position vector of the nth atom at time *t* (i.e., the trajectory obtained from the MD simulation), and ${\langle \cdots \rangle }_{t,\Omega }$ denotes an average over time *t* and orientation Ω. The $I^{\mathrm{inc}}\left(Q,\tau \right)$, calculated by Sassena, is normalized to $I^{\mathrm{norm}}\left(Q,\tau \right)$ as per the normalization procedure described in Appendix C.

$I^{\mathrm{norm}}\left(Q,\tau \right)$ was also calculated with LiquidLib with the same input trajectories as in the calculation with Sassena. The normalization is implicit within LiquidLib. However, the *Q* values allowed to be used in LiquidLib are $2\pi /L\cdot \sqrt{n_{1}^{2}+n_{2}^{2}+n_{3}^{2}}$ with the simulation box length *L* and non-negative integers $\left\{n_{i}\right\}$. Due to this restriction, the *Q* values used by LiquidLib do not match the values used by Sassena exactly. In order to compare LiquidLib and Sassena, the values that are closest to the selected round *Q* values are used. The average over reference time *t* cannot be carried out due to some interdependence between input parameters.

The incoherent intermediate scattering function $I^{\mathrm{norm}}\left(Q,\tau \right)$ is related to the self-diffusion coefficient [59,60], and shows how strongly the positions of the atoms with a time τ between them are correlated with each other. Since for τ=0 no structural changes can happen, the system is perfectly correlated to the reference, so $I^{\mathrm{norm}}\left(Q,\tau \right)$ always starts from 1. If long-range motions are present in the sample, the correlation will be completely lost over time, and the intermediate scattering function decays to 0 because it finds no correlation any more. The speed of this correlation decay depends on the length scale under investigation: the larger the length scale (hence the lower the *Q*), the longer the atoms need to cover the distance, and correspondingly, the slower the decay.

Figure 1 shows the neutron incoherent intermediate scattering function of H_2_O. The scattering patterns, calculated by Sassena and LiquidLib, are compared to data reconstructed from the literature [55] as described in Appendix D. It can be seen that the correlations decay exactly in the ranges of τ and *Q* probed by the simulation and the scattering experiment. The $I^{\mathrm{norm}}\left(Q,\tau \right)$, calculated by Sassena and LiquidLib, is qualitatively similar to its experimental counterpart but decays faster for all *Q* values. The slight mismatch at lower *Q* values between the $I^{\mathrm{norm}}\left(Q,\tau \right)$, calculated by LiquidLib and Sassena (clearly visible around Q = 0.1 Å^−1^), is in agreement with the general behavior of $I^{\mathrm{norm}}\left(Q,\tau \right)$, showing a faster decay with larger *Q*. The higher noise of the LiquidLib result (still far below the scatter of typical experimental data) could certainly be improved by averaging over reference time *t*.

### 2.2. Structure as Seen by Diffraction

The structure can be determined by *diffraction* experiments, where the scattered X-ray/neutron intensity or diffractogram is recorded as a function of momentum transfer only. The typically probed length scales range from some Ångströms to hundreds of nanometers. It is customary to distinguish between two regimes: wide-angle scattering (Q≳1 Å^−1^ correspondingly mainly sensitive to distances r≲6 Å) and small-angle scattering (Q≪1 Å^−1^, correspondingly mainly sensitive to distances ). Since the small-angle scattering signal’s sensitivity is limited to large structures, where always many atoms contribute, a useful concept for small-angle scattering is the local *scattering length density* (SLD, ρ), which is the product of the average scattering length of a group of atoms with their local number density. It is similar in concept to the refractive index in optics [61]. Small-angle scattering will only be caused if there is a *contrast*, i.e., a difference in SLD, between two regions in the sample. To compare the simulations to these experiments, the scattering function S(Q) is calculated from the atomic coordinates obtained from the MD simulation, which can be expressed as
(2)$$S\left(Q\right)={\langle \sum_{j=1}^{N}\sum_{k=1}^{N}b_{j}^{\mathrm{coh},†}b_{k}^{\mathrm{coh}}\exp\left[i\vec{Q}\cdot \left({\vec{R}}_{j}\left(t\right)-{\vec{R}}_{k}\left(t\right)\right)\right]\rangle }_{t,\Omega }\;,$$
where $b_{n}^{\mathrm{coh}}$ is the coherent scattering length for the nth atom and ^†^ denotes the complex conjugate. This can be equivalently formulated using the fact that the scattered intensity is the absolute value squared of the scattering amplitude $F(\vec{Q})$
(3)$$S\left(Q\right)={\langle {\left|F(\vec{Q})\right|}^{2}\rangle }_{t,\Omega }={\langle {\left|\sum_{j=1}^{N}b_{j}^{\mathrm{coh}}e^{i\vec{Q}\cdot {\vec{R}}_{j}\left(t\right)}\right|}^{2}\rangle }_{t,\Omega }\;.$$

The computation results are put on an absolute scale by the normalization procedures described in Appendix C to obtain a diffractogram D(Q).

A naïve calculation of diffractograms from an MD trajectory exhibits a spurious small-angle scattering signal caused by a finite size effect arising from the finite size of the simulation box [52,53]. It appears due to the difference in sample size between an experiment and its corresponding simulation as visualized in Figure 2. In an experiment, the size of the container of the solvent is many orders of magnitude larger than the solute, which is in the nanometer size range. Therefore, the experimental small-angle scattering data show the size and shape of the solute only. For computational efficiency, the size of a simulation box is preferably slightly larger but comparable to the solute; it is therefore no surprise that the discontinuity of density at the edges of the box introduces scattering features in a *Q* range similar to that where the solute itself scatters.

One straightforward way to remove this artifact is to use a bigger simulation box because that would push the spurious signal to a region of smaller *Q* values. However, this approach comes at the cost of even greater computation times. Therefore, two methods to remove this artifact, without increasing the box size, are presented in the following: one, denoted here as *r*-clean (Section 2.2.1), was already implemented in the original version of Sassena. This method solves the problems in the small-angle region but simultaneously results in a wrong wide-angle diffraction pattern. An alternative is the *Q*-clean method that we present here for the first time in Section 2.2.2, which does not suffer from this problem as will be demonstrated in several examples in Section 2.2.3.

#### 2.2.1. *r*-Clean: Subtracting the Average Scattering Length Density of the Solvent from the Scattering Lengths to Remove the Contrast between the Simulation Box and Surrounding Vacuum

This method is based on the fact that, while wide-angle scattering is determined by the scattering lengths of individual atoms, small-angle scattering is only caused by variations in the scattering length density. Therefore, a computation of the small-angle diffraction pattern shows not only the features due to the contrast between the solute and solvent but also the features caused by the contrast between the simulation box and the vacuum around it. The *r*-clean method sketched in Figure 3 operates in the real space, subtracting a suitable value from the scattering lengths of all the atoms within the simulation so that the average scattering length density of the solvent becomes zero. As a result, there is no contrast anymore between the solvent in the box and the vacuum—the spurious small-angle scattering signal disappears.

For this procedure to work, the SLD of the solvent has to be calculated from the scattering lengths—and the to-be-subtracted SLD has to be converted back to scattering lengths that will then be deducted from those of the atoms. In Sassena, the average SLD of the solvent $\rho_{sol}$ is computed by considering atoms as spheres with a radius equal to their tabulated van der Waals radius [62] to be
(4)$${\bar{\rho }}_{\mathrm{sol}}=\left.\sum_{n=1}^{N_{\mathrm{sol}}}b_{n}^{\mathrm{coh}}/\sum_{n=1}^{N_{\mathrm{sol}}}V_{n}^{\mathrm{vdW}}\right.\;,$$
where $N_{\mathrm{sol}}$ is the total number of solvent atoms in the system, $b_{n}^{\mathrm{coh}}$ is the coherent scattering length of the nth atom, and $V_{n}^{\mathrm{vdW}}=\frac{4}{3}\pi {\left(r_{n}^{\mathrm{vdW}}\right)}^{3}$ is the van der Waals volume depending on the van der Waals radius $r_{n}^{\mathrm{vdW}}$ of the nth atom. The same relation is used to obtain a scattering length back from this SLD for each atom in the system so that the corresponding scattering function $S^{r-\mathrm{clean}}\left(Q\right)$ is then calculated as
(5)$$S^{r-\mathrm{clean}}\left(Q\right)={\langle \sum_{j=1}^{N}\sum_{k=1}^{N}\left(b_{j}^{\mathrm{coh}}-{\bar{\rho }}_{\mathrm{sol}}\cdot V_{\mathrm{vdW},j}\right)\left(b_{k}^{\mathrm{coh}}-{\bar{\rho }}_{\mathrm{sol}}\cdot V_{\mathrm{vdW},k}\right)e^{i\vec{Q}\cdot \left({\vec{R}}_{j}\left(t\right)-{\vec{R}}_{k}\left(t\right)\right)}\rangle }_{t,\Omega }\;.$$

The drawback of this approach is that the wide-angle scattering pattern will be completely distorted. This can be understood in the rather extreme case of Figure 3, where the solvent is completely masked, and no solvent-solvent correlations will be visible anymore in the wide-angle scattering pattern. In general, scattering signals will not simply disappear, but a wrong scattering pattern (namely one belonging to a substance with completely different scattering lengths) will emerge as will be shown in an example in Section 2.2.3.

#### 2.2.2. *Q*-Clean: Subtracting the Scattering Amplitude of the Simulation Box in Reciprocal Space

We introduce here a new method that allows dealing with the finite size effect without introducing artifacts in the wide-angle scattering region that will be called *Q*-clean. As summarized in Figure 4, the scattering amplitude $F^{\mathrm{tot}}\left(\vec{Q}\right)$ of the whole system is calculated as in the naïve approach. Before converting the amplitude to a scattering intensity and performing the orientational average, the scattering amplitude of a cuboid with the same size as the simulation box and a homogeneous SLD is subtracted. This theoretical cuboid has no atomic structure; it therefore creates no wide-angle scattering at all. When using the average SLD of the solvent ${\bar{\rho }}_{\mathrm{sol}}$ for this cuboid, only the small-angle scattering of the solute is visible after subtraction.

The scattering amplitude of the total system, i.e., solute and solvent, $F^{\mathrm{tot}}\left(\vec{Q}\right)$ (not the scattering intensity $S(\vec{Q})$) was calculated as specified in Equation (Equation 3), as was the scattering amplitude of a homogeneously filled cuboid: (6)$$\begin{array}{cc} F^{\mathrm{sol}}\left(\vec{Q}\right) & =e^{i\vec{Q}\cdot {\vec{R}}_{\mathrm{box}-\mathrm{centre}}}\cdot {\;\int \int \int \;}_{V}{\bar{\rho }}_{\mathrm{sol}}e^{i\vec{Q}\cdot \vec{R}}dV\\ \; & =e^{i\vec{Q}\cdot {\vec{R}}_{\mathrm{box}-\mathrm{centre}}}\cdot {\bar{\rho }}_{\mathrm{sol}}\cdot abc\cdot \mathrm{sinc}\left(Q_{x}a/2\right)\mathrm{sinc}\left(Q_{y}b/2\right)\mathrm{sinc}\left(Q_{z}c/2\right)\;, \end{array}$$
where ${\vec{R}}_{\mathrm{box}-\mathrm{center}}$ is the position vector of the center of the simulation box, $\vec{Q}$ is the scattering vector with Cartesian components $\left(Q_{x},Q_{y},Q_{z}\right)$, $\vec{R}$ is the position vector, and *V* is the volume domain of the cuboid with side lengths *a*, *b*, and *c*. The corresponding scattering function is then
(7)$$S^{Q-\mathrm{clean}}\left(Q\right)={\langle {\left|F^{\mathrm{tot}}\left(\vec{Q}\right)-F^{\mathrm{sol}}\left(\vec{Q}\right)\right|}^{2}\rangle }_{t,\Omega }\;.$$

This procedure removes the scattering signal caused by the contrast between the SLD of the simulation box and the vacuum around it. What remains is the small-angle scattering signal of the simulation box content, i.e., the contrast between solute and solvent, and the wide-angle scattering signal of all atoms in the simulated system. The intensity ratio between these two regimes is therefore governed by the ratio between the number of solute and solvent molecules—it assumes that the concentration of the solute in the simulated volume is the same as in the experiment.

#### 2.2.3. Calculation of Example Diffractograms

The application of the *r*-clean and *Q*-clean methods in Sassena in comparison to the naïve approach in Sassena and LiquidLib is demonstrated in this section. Since neutrons can distinguish between isotopes, different mixtures of normal water and heavy water, named H_2_O, D_2_O, HDO, and the null mixture (cf. Table 1), were considered for this purpose. The mixing ratio of the null mixture is chosen such that the average scattering length of hydrogen/deuterium is zero and therefore only oxygen–oxygen correlations determine the scattering signal.

Unlike the incoherent intermediate scattering function, calculations related to diffractograms require the preservation of structure (i.e., correct density and interatomic distance) in the trajectory. Therefore, output trajectories were saved in *wrapped* format, which restricts the coordinates to within a central periodic image. The unrealistic jumps of atoms around the periodic boundary, enforced due to wrapping, are not a source of any error for structure investigation. The small-angle and wide-angle diffraction patterns were calculated from simulated trajectories using naïve, *r*-clean, and *Q*-clean methods within Sassena. The diffractogram, calculated using LiquidLib, was limited by Q≥2π/L, and therefore, the diffractogram in the small angle could not be computed. LiquidLib used the same input trajectories as Sassena. All results are compared with the experimental results in Figure 5.

Water does not exhibit inhomogeneities on the nanometer length scale, and therefore, the experimental small-angle diffraction patterns do not exhibit any peaks, which is clearly visible even though the data are subject to systematic and statistical errors. In contrast, the naïvely calculated pattern using Sassena shows clear features due to the finite size effect. This problem is solved in LiquidLib by not calculating anything in this *Q* range. Sassena solved this problem by removing this spurious signal using the *r*-clean method, but this altered the wide-angle diffractograms, which then deviated clearly from the experimental data. This is particularly well visible in the case of D_2_O, but all curves exhibit deviations—some peaks get smaller, some get bigger. Applying the new *Q*-clean method instead, Sassena removes the spurious signal in the small-angle region without changing anything in the wide-angle region.

The diffractogram calculated by LiquidLib also suffered from a few other differences in comparison to Sassena, apart from the limitation in the choice of *Q* values: (a) the average over time *t* was performed over a smaller number of frames; and (b) the orientational average was performed over fewer *Q* values in comparison to Sassena. Despite these differences, the diffractograms calculated by LiquidLib match quite well with the naïve and *Q*-clean calculations of Sassena. A few sporadic points in the case of D_2_O and HDO are well within the error bar of experiments.

Switching to cases where a small-angle scattering signal has features, the two subtraction methods were applied to a toy example of a spherical solute in solvent as shown in Figure 6a. The small-angle diffraction pattern for the spherical solute only was calculated separately along with the analytical diffractogram of a sphere as the ground truth. The diffraction pattern calculated from the solute and solvent was expected to match these two curves completely, after the removal of the finite size effect. The *Q*-clean method performed as well as the *r*-clean and removed the finite size effect, which is seen in the diffraction pattern calculated with the naïve method.

The final example is a lysozyme protein immersed in water, shown in Figure 6b. This example was chosen to demonstrate the performance of the *Q*-clean method in a real-life example. The calculated results are compared with the experiments [64]. Both *r*-clean and *Q*-clean produced a good match with the experimental data and removed the finite size effect seen in the naïvely calculated pattern. The quality of match between the experiment and calculated pattern deteriorates slightly in the region where *Q* is greater than 0.1 Å^−1^. At a *Q* higher than 0.2 Å^−1^, the experimental data suffer from statistical error [64] and incoherent scattering, which was not included in the calculations. The difference between the *r*-clean and *Q*-clean results at *Q* values above 0.1 Å^−1^ will be discussed later.

### 2.3. Increasing the Computation Speed

The issue of computation speed is addressed separately for coherent and incoherent calculations due to algorithmic differences, described in Appendix A. Before attempting to increase the speed of the process, an analysis was carried out to identify if some parameters of the simulation had a larger impact than others. The main factors influencing the computation time are the number of atoms, the length of the trajectory, i.e., the number of time frames, and the number of evaluated scattering vectors $\vec{Q}$, which in the case of orientational averaging is the product of the number of *Q* lengths and the number of orientations [39]. The effect of variation of each of these parameters on the computation time was studied separately for coherent and incoherent scattering. Only one parameter was varied at a time, and all other parameters remained at the value of the *base configuration*. Table 2 shows how each of the parameters was varied.

Figure 7 shows the scaling of the computation time for the optimized version of Sassena created in the frame of this work. The investigation was performed on the in-house cluster described in Section 4, with 12 MPI processes and 2 OpenMP processes.

It can be seen that the computation time scales approximately linearly with all parameters, barring the exception of the number of frames when calculating incoherent spectroscopy. The coherent type of calculation for the base configuration took just over 12 h on one core as shown in Figure 8. Extrapolating this value, it would take around 25 years to calculate the scattering of a simulation box with a side length of only 100 nm, containing an estimated 1 billion atoms.

This value visualizes the need for efficient and fast calculations. In the following, the single-core performance and then the multi-core performance related to parallelism are analyzed and optimized to reach this goal.

#### 2.3.1. Single-Core Performance: Efficient Vectorization

Modern processors work with *vectorization*: Each register can perform the same operation simultaneously on several data, which is referred to as SIMD [65]. How much data can be processed simultaneously depends on the required precision of the data and the available instruction set. The precision of the data decides the length of the data related to one number. In the current case, calculations are performed in double precision (64 bits). The instruction sets decide the length of the register. In our system, the available instruction sets were SSE2, AVX2, and AVX-512 as mentioned in Table 3. Using SSE2 (128 bits), two numbers can be processed simultaneously; this goes up to four numbers with AVX2 (256 bits) and eight numbers with AVX-512 (512 bits) [66,67].

The original authors of Sassena already recognized the possibility of enhancing single-core performance [39] but did not publish a detailed performance analysis. Profiling the program with Intel Advisor [68] showed that the loops were not vectorized in the original Sassena when compiled with the GNU MPI compiler [69]. We therefore decided to change to the Intel MPI compiler [70] that enforced auto-vectorization and the use of the Intel Math Kernel Library [71], resulting in a significant speed increase shown in Figure 8. Care was taken to ensure that the compiler uses the best instruction set available on the processor, further increasing speed at the cost of reduced portability of the binary. In the figure, it can be seen that nearly a factor of two was gained for incoherent scattering, and approximately a factor of eight was gained for coherent scattering. A convenient option for the users to choose their preferred compiler was added to the compilation scripts of Sassena.

#### 2.3.2. Multi-Core Performance: Introduction of OpenMP

The multi-core performance thrives on efficient sharing of the workload, for which two approaches are available: distributed and shared memory parallel computing. Distributed memory parallel computing, often implemented using MPI, distributes the data to the memory of the individual cores, which then work on their share of the data and feed their respective results back together at the end. Shared memory parallel computing, often implemented using threading or OpenMP, uses a common memory which is then shared by all the participating cores. Each approach has its share of advantages and disadvantages. In order to take full advantage of both, the implementation of hybrid parallelism (i.e., the simultaneous use of distributed and shared memory parallel computing) in Sassena is attempted here.

The original version of Sassena already used distributed and shared memory parallel computing, which was implemented through MPI [72] and threading [73], respectively. The threading implementation targets only very specific use cases where MPI parallelism does not perform well. In an attempt to introduce hybrid parallelism, the use of OpenMP [74] was explored and implemented for the inner two loops of the calculation of coherent scattering patterns. The computation times with the base configuration for the Sassena compiled with Intel MPI compiler were recorded at first with 24 MPI processes and then with 24 OpenMP processes for the calculation of coherent scattering, and 24 MPI processes for the calculation of incoherent scattering. The results are shown in Figure 8: up to two orders of magnitude speed increase were achieved in comparison to the GNU compiler version run on one core, but the OpenMP implementation does not outperform the MPI implementation.

Apart from the computation time, scalability was also investigated: the ratio of the computation time when running the program on one core to the computation time when running the program on *N* cores. Scalability shows the efficiency of the parallelization: Ideally, one would hope for an algorithm to speed up *N* times when run on *N* cores compared to the time it takes to be run on one core. In reality, there will be more and more inefficiencies that will make the speedup sub-linear [75,76]. Figure 9 shows the scalability of Sassena with MPI and OpenMP for coherent scattering and the scalability of Sassena with MPI for incoherent scattering. It can be seen that the scaling of computations for coherent and incoherent measurements is very similar. The new OpenMP implementation scales only slightly less well than the original MPI implementation. The analysis of computation time vs. the number of cores is also provided in Lindner et al. [39] with thousands of cores, which shows Sassena’s applicability to different computational environments.

Through the introduction of optimizations related to the single-core and multi-core performance, Sassena is now enabled to handle MD simulations with approximately one order of magnitude more atoms or longer trajectories than before in a reasonable time. This is a necessary step towards the calculation of scattering patterns for the large simulations of systems.

## 3. Discussion

### 3.1. Dynamics—Atomic

The three important contributions to the dynamics, which are measured by the quasielastic spectroscopy experiment, are long-range diffusion, reorientation of the molecules, and fast vibrations [55,60]. The TIP3P/flex force field [77,78], used in this work, can reproduce the reorientation phenomenon well compared to the experimental observation [79] but overestimates the self-diffusion coefficient [59]. Due to this overestimation, the intermediate scattering function shows faster dynamics than the experimental data as can be seen in the quicker decay of the intermediate scattering functions in Figure 1. One of the best estimations of the self-diffusion coefficient [60] and a slower decay in the intermediate scattering function [14] can be achieved by the 4-site rigid model TIP4P/2005, but it cannot estimate the reorientation phenomenon as well as the TIP3P/flex force field [60]. In reality, no force field can reproduce all the properties of water in an MD simulation [14,60,77].

An open point for the calculation of spectroscopic patterns, in general, is the calculation of the coherent part of dynamics. The total intermediate scattering function measured in an experiment contains contributions from both incoherent and coherent parts. Unlike the incoherent part, which is not sensitive to structure, the coherent part is sensitive to both dynamics and structure. To incorporate the proper dynamics, the saved trajectory cannot be stored in a wrapped format because it causes nonphysical motions of atoms across the boundary as highlighted in Section 2.1. However, the calculation of the coherent scattering signal requires this wrapping because it preserves the correct density and therefore the structure. Due to this dilemma, we do not see how the coherent part of the intermediate scattering function could be calculated adequately. However, the calculation of the dynamics of water presented in this contribution is not affected by this problem because the incoherent scattering dominates the scattering signal due to the much larger incoherent (compared to coherent) scattering cross section of hydrogen.

In order to compare the performance of Sassena with similar software, the incoherent intermediate scattering pattern was also calculated using LiquidLib. The comparison between LiquidLib and Sassena confirms the overall trend of the quicker decay of I(Q,τ) with respect to τ at higher *Q*. The restriction of the *Q* values with which LiquidLib computed the scattering patterns is due to the reciprocal lattice approach [80]. It chooses points where no contribution from the shape and size of the simulation box is visible. In contrast, Sassena can calculate scattering patterns at all *Q* values. Since the incoherent calculations are not influenced by the shape and size of the simulation box, the freedom in the choice of *Q* values is an advantage and allows better benchmarking with experimental results. The reciprocal lattice approach also limits the number of orientations over which the orientational average can be performed, but it does not leave any significant impact in the case of water due to its isotropic structure. LiquidLib also could not perform the average over reference time *t*, which signifies the ensemble average. This is the reason behind the noise at a higher correlation time τ because the size of the ensemble of states increases at a higher τ. In contrast, the number of frames related to the ensemble average is not dependent on any other input parameter in Sassena. This allows a noise-free calculation, especially at higher τ.

### 3.2. Structure—Atomic and Nanoscopic

Diffractograms, calculated by Sassena and LiquidLib, could be compared one to one, despite the restriction in the choice of *Q* values within LiquidLib as shown in Figure 5. In addition to this, the ensemble average could be performed for LiquidLib and Sassena both in the case of diffractograms. Although the number of frames over which the average was performed in LiquidLib was smaller than in Sassena, it did not leave a very large impact. No control over the orientational average, due to the restriction in the choice of *Q* values, persisted for diffractograms as well. This might have caused one to two sporadic points in the diffractogram of D_2_O and HDO.

A naïve calculation of diffractograms by Sassena and LiquidLib from the simulated atomic structure of different mixtures of H_2_O and D_2_O show fairly good agreement with the experimental data in Figure 5 in the wide-angle region. There are smaller differences, like a too-strong oscillation of the calculated patterns at high *Q*, best seen in the case of D_2_O, and a mismatch at intermediate *Q* values (∼1–5 Å^−1^), which highlight the inaccuracies of the employed force field: a too-well-defined O−H bond length, and differences between the model and reality in the intermolecular correlations, respectively. This information can be used to improve the inter-atomic potentials or their parameters [14].

The biggest disagreement between the naïvely calculated scattering curves and experiments can be observed in the small-angle region of the diffractogram due to the finite size effect [51,52,53,54]. This has nothing to do with the MD simulation, only the scattering pattern calculation. This kind of finite size effect can be compared to the peak broadening due to the finite size of grain in a crystalline sample, where the ideal scenario would be an infinite crystal [81,82]. LiquidLib solves this problem by restricting its calculation to Q≥2π/L, where *L* is the length of the simulation box. The restriction would necessitate a bigger *L* for the calculation at smaller *Q* values. In contrast, the *r*-clean and *Q*-clean methods, employed in Sassena, remove these strong features eliminating the need for a bigger simulation box. In between the *r*-clean and *Q*-clean methods, the *Q*-clean method has the clear advantage over *r*-clean because it performs equally well at low *Q* while keeping the scattering pattern at high *Q* undisturbed, as shown in Figure 5.

Also at low *Q*, the *Q*-clean performs better than the *r*-clean method since it does not introduce a shift in the computed D(Q): the calculation is directly comparable to the experimental data. This shift is not visible in Figure 6 because it was manually corrected by multiplying the *r*-clean result with an empirically determined factor to ease the visual comparison to the *Q*-clean method. The case of lysozyme shows, however, that a qualitative difference between the two methods starts already in the *Q*-range covered by the small-angle scattering data. From this comparison alone, it is not possible to decide if the *r*-clean or *Q*-clean method is closer to the ground truth: the comparison to the experimental data suffers from the fact that (a) the employed force field was not particularly good, and (b) incoherent scattering was not taken into consideration in the calculations. What can be concluded is that at least one of the computed scattering curves (both using the same simulation trajectory) is distorted to a degree that should not be neglected; the results of Figure 5 show that the *r*-clean method is affected by distortions, while the *Q*-clean is not.

The calculation of small-angle scattering from the spherical solute in a solvent toy example (Figure 6) shows that our method predicts exact results. This strengthens our argument that the source of the mismatch observed for lysozyme is not due to our method but rather the simulation, which did not capture the full complexity of the system.

The spurious small-angle scattering is only one type of finite size effect. Others, not corrected by the methods presented here, arise when correlations between atoms still persist across distances on the same order of magnitude as the size of the simulation box [83]. These might affect the simulation itself (e.g., insufficient sampling of the ensemble [52]) or the scattering pattern calculation (e.g., Bragg peak broadening of a crystalline material [82,84]). The *Q*-clean method introduced here is currently only implemented for cubic simulation boxes; a future extension could incorporate other shapes of simulation regions like octahedrons or dodecahedrons [85].

It should be noted that the investigation of the atomic structure of crystals is very sensitive to the numerically performed orientational average: since the Bragg peaks are very sharp in reciprocal space, a very large number of $\vec{Q}$ vectors have to be computed to ensure representative sampling of the scattered intensities. For structural calculations, this can be circumvented by implementing the analytical solution for the orientational average [86,87,88] as performed, for example, by the program Debyer [89] (not maintained anymore).

### 3.3. Increasing the Computation Speed

The question related to computation speed was split into two parts: single-core performance and multi-core performance.

The single-core performance mainly benefited from a change of compiler, which enforced vectorization. The used processor was equipped with the instruction sets SSE2, AVX2, and AVX-512, which are expected to give a theoretical speed boost of two, four, and eight times, respectively. Since it was ensured through a compiler flag that the best instruction set of the system would be used, a theoretical maximum of an eight-times speed boost was expected. The results for the calculation of coherent diffraction indeed did show a speed boost of eight times. A computation rarely achieves the theoretical optimum because there is always a part of the code that is not vectorized [75]. We hypothesize that this speed increase is due to the additional contribution from the efficient Intel Math Kernel Library [71].

The calculation of incoherent spectroscopy did not benefit equally from the vectorization and use of the Intel Math Kernel Library. This indicates that the bottlenecks are not the non-vectorized loops. Since the calculation of incoherent spectroscopy computes the autocorrelation function, which is implemented via the FFTW algorithm in Sassena, this is a likely place to concentrate on for further optimizations.

It is noteworthy that the single-core optimizations presented here can only be accessed when using the Intel MPI compilers. However, the GNU MPI compiler offers vectorization as well. Future development could therefore aim to enable vectorization also while compiling with the GNU MPI compiler through, for example, compiler flags or pragmas to allow the user a free choice of compiler.

Generally speaking, multi-core optimization with distributed memory parallelization alone suffers from high memory usage and overhead. On the other hand, shared memory parallelization has limited capacity in comparison to distributed memory parallelization [90] and can suffer from load balancing issues [91]. Therefore, the main goal of the multi-core optimization was to implement hybrid parallelization, i.e., distributed memory parallelization (MPI) and shared memory parallelization together.

The existing implementation of shared memory parallelization using threading in the original version of Sassena was only useful for very specific cases. It ensures that Sassena can parallelize well for cases where MPI performs poorly. Since MPI and shared memory parallelization should be equally effective to ensure efficient hybrid parallelization, an additional layer of shared memory parallelization was added using OpenMP. Although the scalability with OpenMP and MPI was comparable, an improvement over the MPI implementation could not be achieved with hybrid parallelization in this work. However, finding a perfect combination of MPI and OpenMP is a cumbersome problem in many cases [90].

The computation speed of Sassena was also compared with LiquidLib as mentioned in Figure 8. For coherent calculations, Sassena compiled with the Intel compiler was approximately 18 times faster than LiquidLib compiled with the GNU compiler with 24 MPI processes, despite considering more frames for ensemble averaging in the case of Sassena. For incoherent calculations, the GNU-compiled LiquidLib was about as fast as the Intel-compiled Sassena when both were run with one MPI process. However, in this calculation, LiquidLib did not perform an ensemble average. This caused noise for the calculation of the incoherent intermediate scattering pattern as shown in Figure 1, and more importantly, it restricted the number of allowed MPI processes to a maximum of 1 for LiquidLib. This means that Sassena can easily outperform LiquidLib by using more MPI processes. It is possible to include the ensemble average in LiquidLib, allowing MPI parallelism for incoherent calculation at the cost of reduced time points. However, it was not attempted due to technical difficulties in the implementation.

## 4. Materials and Methods

A summary of technical details regarding all simulations, the calculation of the scattering patterns from them, and relevant experiments are described in this section. All computations were performed on our in-house cluster, characterized in Table 3.

### 4.1. Simulations

The output of all simulations includes a pdb file of the initial structure, providing information about which atom is which element, and a dcd trajectory file, providing information about the atomic coordinates as a function of time. In general, dcd files are saved in wrapped format for the investigation of structure and in unwrapped format for the investigation of dynamics.

#### 4.1.1. Water

A detailed description of the MD simulation of water is given in Appendix B.

#### 4.1.2. Spherical Solute in Solvent

The atomic coordinates were generated for 1 timestep directly from the coordinate generation script [92,93] without the use of an MD engine and saved as pdb and dcd files. “Atoms”, either assigned to be solute or solvent material, were placed on a simple cubic lattice with a unit cell edge length of 1 Å. A 2D cut through the middle of this 3D structure is shown in Figure 10. In total, the system contains 8000 atoms, which are uniformly distributed in a cube of side length 20 Å. The coordinate of the atom closest to the origin is (0.5, 0.5, 0.5); the whole assembly is positioned in the first octant of the coordinate system.

#### 4.1.3. Lysozyme in Water

The details of the MD simulation of lysozyme in water can be found in Appendix E.

### 4.2. Calculation of Scattering Curves from Simulations

For calculation with Sassena, pdb and dcd files are used as input from the simulation. For the calculation with LiquidLib, the xyz file format is generated from pdb and dcd files using the program Visual Molecular Dynamics (VMD) [45]. In the case of Sassena, the values of the scattering lengths are read from a database, which is shipped along with Sassena. In contrast, LquidLib requires scattering lengths to be mentioned explicitly in the input script. The values can be found in literature [56].

#### 4.2.1. Probing the Dynamics

##### Water

In the case of Sassena, the incoherent intermediate scattering function of water was calculated using Sassena for 20 uniformly spaced values of *Q* from 0.1 Å^−1^ to 2.0 Å^−1^. The orientational average was carried out explicitly by a Monte Carlo scheme, computing the scattering functions for 10 orientations of $\vec{Q}$ and averaging over the results (for a detailed description of Sassena’s algorithm, refer to Appendix A).

In the case of LiquidLib, an input file specified the value of the start frame, end frame, and frame interval as 0, 10,001, and 1, respectively. The values of the trajectory delta time, time interval, and number of time points were set to 0.04, 1, and 10,001 with a linear time scale type. The value of the number of frame intervals is dependent on the above parameters and could only be set to a maximum value of 1. Although the range of *k* was set from k=0 to k=2 with an interval of 0.1, where *k* is an alias of *Q*, the values of *k* at which the scattering patterns were calculated were restricted by the length of the simulation box, which was given as 46.584 Å.

#### 4.2.2. Probing the Structure

##### Water

In the case of Sassena, the wide-angle diffraction pattern of water was calculated at 601 uniformly spaced points with 0 Å^−1^≤Q≤30 Å^−1^. For the small-angle diffraction pattern, 601 uniformly spaced points were considered in the range 0 ≤Q≤30 Å^−1^≤Q≤1 Å^−1^. In both cases, the orientational average was carried out over 100 directions of $\vec{Q}$.

In the case of LiquidLib, the values of the start frame, end frame, and frame interval were set to 0, 10,001, and 1, respectively. The value of the number of frame intervals could not be set to 10,001, like Sassena. We set it as 3999. The range of *k* (alias of *Q*) was set as k=0 to k=30 with an interval of 0.05. The length of the simulation box, governing the actual values of *k*, was given as 46.61 Å. For the sake of visual ease, every fifth value of *Q*, starting from the first, is plotted in this contribution. In order to be consistent with the Sassena result, the LiquidLib results were multiplied by $N{\langle b^{\mathrm{coh}}\rangle }^{2}$, where *N* is the number of atoms and $\langle b^{\mathrm{coh}}\rangle$ is the average coherent scattering length, before applying the normalization.

For all mixtures of heavy water D_2_O and normal water H_2_O, the MD trajectory of H_2_O was used as an input, and the coherent scattering length of hydrogen $b_{H}^{\mathrm{coh}}$ was adjusted as per Table 1. This approach works here because H_2_O and D_2_O have in very good approximation the same molecular structure albeit different dynamics.

In order to remove spurious signal in the small-angle region of the computed curves, the *r*-clean was triggered by selecting all atoms in the background.factor.selection field of Sassena. The newly implemented *Q*-clean method was triggered by specifying a cube of edge length 46.61 Å, center at (0,0,0), and the sample-corresponding value of the SLD as listed in Table 1 in Sassena’s background.cut.box field.

##### Spherical Solute in Solvent

The small-angle diffraction of the whole system was calculated using Sassena for 301 uniformly spaced points with 0 Å^−1^≤Q≤1 Å^−1^. The orientational average was taken over 100 randomly distributed directions. The coherent scattering length of solvent and solute atoms were taken as 6.646 fm and −3.739 fm, respectively.

For the *r*-clean method, the solvent atoms were selected in Sassena’s input script. The resulting change of scattering lengths in the computation led to a change in the intensity of the result, which was counteracted by multiplication with an empirically determined factor of $5.82\times 10^{-3}$. For the *Q*-clean method, a cube of edge length 20 Å, positioned with its center at (10, 10, 10), and a scattering length density of 6.646×10^−5^ Å^−2^ (the solvent SLD) was employed.

##### Lysozyme in Water

The small-angle diffraction of neutrons by lysozyme in H_2_O was calculated using Sassena on 581 uniformly spaced points with 0 Å^−1^ ≤ Q ≤ 1 Å^−1^. The orientational average was carried out over 10 directions.

For the *r*-clean method, the H_2_O molecules were selected in Sassena’s input script, and the shift of the resulting scattering curve was corrected by an empirically determined multiplication factor of 14.29. For the *Q*-clean method, a cube was defined with an edge length of 69.49 Å, center at (34.745, 34.745, 34.745), and the scattering length density of H_2_O (cf. Table 1), −0.0559×10^−5^ Å^−2^.

### 4.3. Experiments

Real experimental data were extracted from the literature. The only exception is the toy example of a spherical solute in solvent, where the result of a computation using only the solute and the analytical result was used as reference.

#### 4.3.1. Water Dynamics

The diffusive motions in water were studied by a quasielastic neutron spectroscopy measurement of H_2_O [55]. The raw data themselves were not available for this experiment; therefore, they were reconstructed using the fit formula and optimal parameters determined in their original analysis (details are given in Appendix D). Measured at the IN6 spectrometer at the Institut Laue Langevin (ILL) with an energy resolution of about 35 μeV (half width at half maximum) and a cut-off of the experimental data at an energy transfer of 3 meV, the data are primarily sensitive to processes happening on a time scale between roughly 1 ps and 100 ps.

#### 4.3.2. Water Structure

The experimental wide-angle neutron diffraction data related to the structure of the water samples defined in Table 1 are taken from a meta analysis of several datasets measured at different instruments [94], where the average of the measurements is given together with their root mean squared deviations as error bars.

#### 4.3.3. Spherical Solute in Solvent

To obtain a reference “data” set, the scattering pattern of only the spherical solute (i.e., after the deletion of all solvent atoms) was calculated with Sassena. As the deletion of scatterers changes the scattering intensity, the result had to be multiplied by an empirically determined factor of 7.71 to be comparable to the calculated scattering curves of the whole system. The analytical formula of form factor the sphere, taken from the literature [95], is also added as a reference “data”. It was also multiplied by a factor of 7.71. The radius of the sphere was 5 Ångstrom, and the scattering length density was −3.941×10^−5^ Å^−^.

#### 4.3.4. Lysozyme in Water Structure

The small-angle neutron diffraction data for lysozyme in H_2_O buffer were taken from the SASDPZ4 entry in the small-angle scattering biological data bank SASDB [96], a curated repository for small-angle scattering data and models. This entry is a consensus dataset of 76 experiments performed at 4 SANS instruments [64]. Since the scattering intensities are given in arbitrary units, they were scaled by an empirically determined factor of 100 to match the computation result.

## 5. Conclusions

This work uses Sassena [42] as a potent tool for the calculation of scattering patterns from MD simulations. We achieved a substantial improvement in the computation speed over the original version of Sassena, which was already very fast. In the future, we expect that this speed gain will enable the fit of force field parameters to scattering data, which requires many iterations of MD simulation and scattering pattern calculation. Together with the proposed method to alleviate a finite size effect visible in the small-angle region, this makes it possible to compute scattering patterns from nanoscopically structured samples and validate them against experimental scattering data. We plan to use this for explaining in situ neutron scattering studies of nanoscopically sized hydrogen storage materials [97].

## Figures and Tables

**Figure 1 ijms-25-01547-f001:**
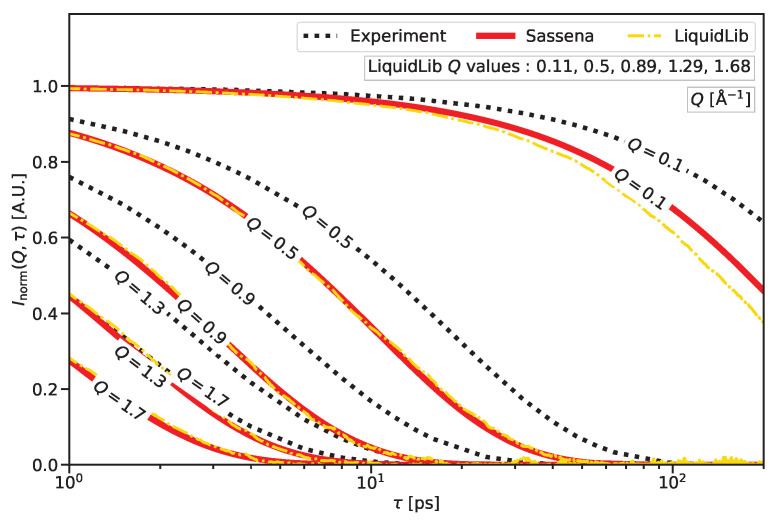
Incoherent intermediate scattering function of H_2_O at 293 K for a range of selected *Q* values. Shown are the experimental results reconstructed from the literature [55] as detailed in Appendix D together with the result of an MD simulation of the TIP3P/flexible water model. The scattering function was calculated with both Sassena and LiquidLib. As LiquidLib restricts the choice of *Q* values, the available values closest to the Sassena values were selected.

**Figure 2 ijms-25-01547-f002:**
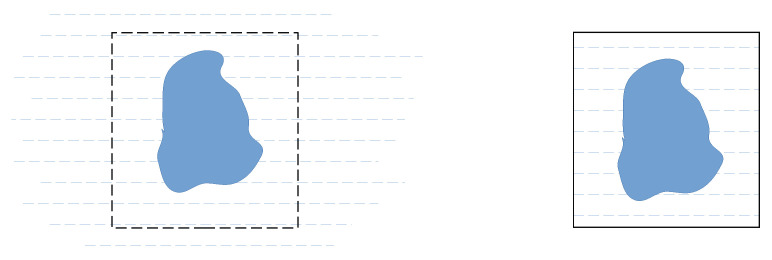
(**Left**): the solute immersed in a solvent during an experiment, where the boundary of the solvent is far away from the solute. The representative region that would be considered for a simulation is shown as a dashed box. (**Right**): a solute immersed in a solvent during the simulation. There is a finite simulation box beyond which there is no solvent, which causes a spurious small-angle scattering signal in the computed scattering curve that is not present in the experimental data due to the much larger sample size.

**Figure 3 ijms-25-01547-f003:**
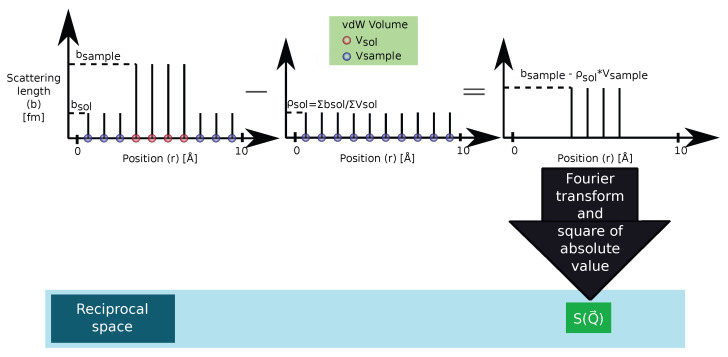
Schematic of the *r*-clean method, where a suitable scattering length is subtracted from each and every atom to remove the contrast between the solvent and the vacuum around it. As a result of this subtraction, the scattering lengths of the sample atoms are changed to contrast with respect to the solvent.

**Figure 4 ijms-25-01547-f004:**
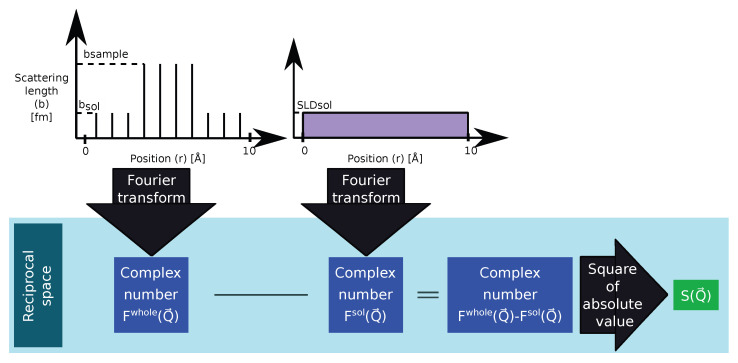
Schematic diagram of the *Q*-clean algorithm. In this method, the scattering from a continuous medium is subtracted from the scattering of the whole system. The subtraction is accomplished by the following steps: Fourier transform of the whole system and the continuous medium to be subtracted, subtraction of complex numbers obtained from the Fourier transforms, and modulus square of the resultant complex number.

**Figure 5 ijms-25-01547-f005:**
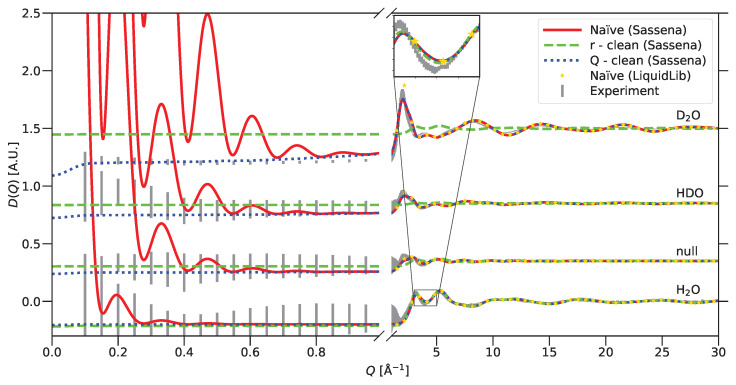
Diffraction pattern of different mixtures of H_2_O and D_2_O: the naïve calculation, after application of the *r*-clean method, and after the application of the *Q*-clean method with Sassena; the naïve calculation with LiquidLib; compared to experimental data.

**Figure 6 ijms-25-01547-f006:**
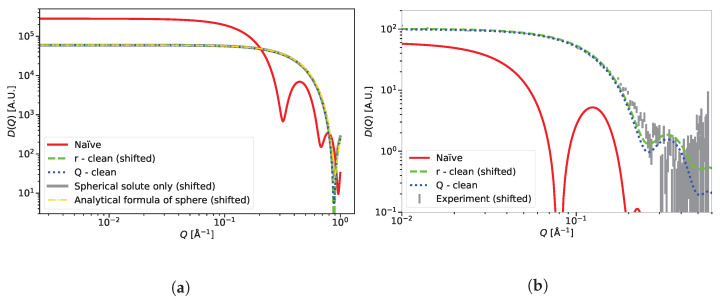
Application of the *r*-clean and the *Q*-clean method on several systems, compared to reference data, calculated analytically and using Sassena and the naïve computation result. (**a**) Diffraction pattern of the toy example of a spherical solute in a solvent; (**b**) Diffraction pattern of lysozyme in water.

**Figure 7 ijms-25-01547-f007:**
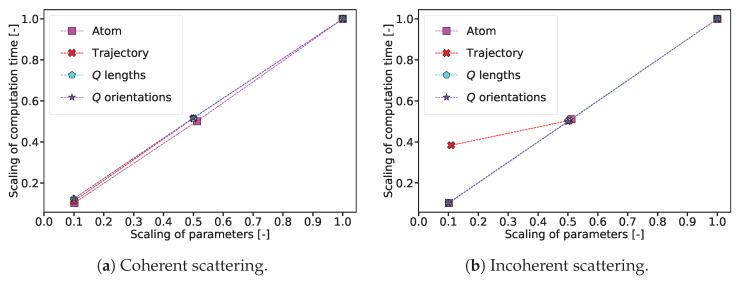
Computation time in dependence of several parameters. For each individual parameter scan, the computation time is shown with respect to the computation time at the base configuration. This process was repeated for each parameter according to the values listed in Table 2, which are shown here normalized to the parameter value at the base configuration.

**Figure 8 ijms-25-01547-f008:**
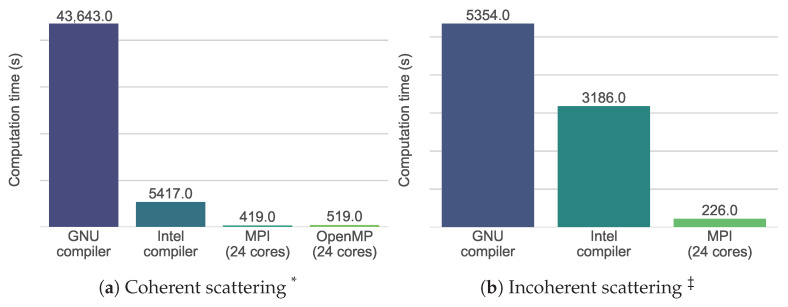
Computation time in seconds for the base configuration (cf. Table 2) using different versions of Sassena run on the in-house cluster described in Section 4. Compared are the original version of Sassena without vectorization (GNU compiler), run on one core; the adapted version presented in this contribution, where vectorization is introduced by the Intel compiler, run on one core; the same version, run on 24 cores using MPI parallelization; and finally for coherent scattering, the same version run on 24 cores using the newly introduced OpenMP parallelization. ^*^ LiquidLib base config (refer Table 2)—coherent calculation took 7742 s with 24 MPI processes. ^‡^ LiquidLib base config (refer Table 2)—incoherent calculation took 3263 s with 1 MPI process only. More than 1 MPI process was not allowed due to interdependency between the parameters.

**Figure 9 ijms-25-01547-f009:**
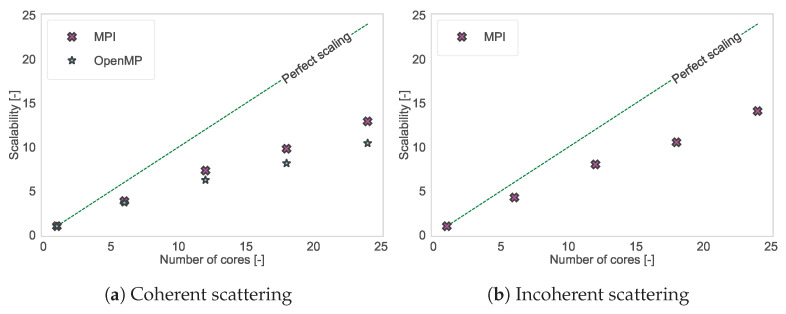
The scalability of the original MPI parallelization and the newly introduced OpenMP parallelization for different kinds of computations with Sassena. Also shown is the line of perfect scalability for comparison.

**Figure 10 ijms-25-01547-f010:**
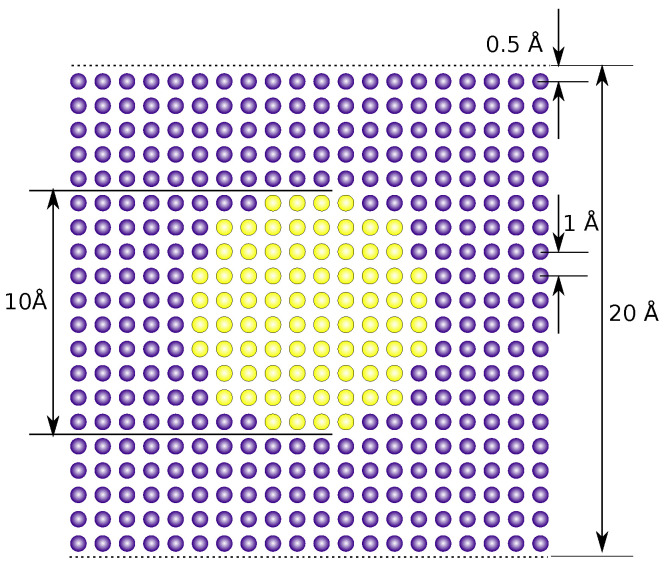
A two-dimensional cut through the middle of the spherical solute in a solvent toy system. The solute atoms are shown in yellow, and the solvent atoms in violet.

**Table 1 ijms-25-01547-t001:** The isotopic composition of the water mixtures used in this contribution. In the *null* sample, scattering from hydrogen and deuterium cancel each other completely. The values given for the scattering length density (SLD) include the contribution by oxygen. It was calculated using the density of H_2_O at 300 K taken from the CRC handbook [63] and assuming a constant atomic number density across all isotopologues.

Name of Sample	Mole Fraction of H_2_O	avg. $b_{H}^{\mathrm{coh}}$ [fm]	SLD [$10^{-5}$ ${Å}^{-2}$]
D_2_O	0.00	6.671	0.6380
HDO	0.50	1.466	0.2912
null	0.64	0.000	0.1934
H_2_O	1.00	−3.739	−0.0559

**Table 2 ijms-25-01547-t002:** Variation of the different parameters influencing the computation time of Sassena: the number of atoms, length of trajectory, number of $\vec{Q}$ orientations, and number of *Q* lengths. Only one parameter was varied at any given time while the others were kept at the value denoted in the column *base configuration*.

Parameter	Coherent		Incoherent	
	Base Configuration *	Parameter Variations	Base Configuration ^‡^	Parameter Variations
Number of atoms	10,125	5184, 1029	10,125	5184, 1029
Length of trajectory	10,001	5001, 1001	101	51, 11
Number of $\vec{Q}$ orientations	100	50, 10	100	50, 10
Number of *Q* lengths	100	50, 10	100	50, 10

* LiquidLib (Base config-coherent)—10,125 atoms, 10,001 frames (3999 frames for ensemble average), 100 *Q* values. Number of $\vec{Q}$ orientations cannot be specified in LiquidLib. ^‡^ LiquidLib (Base config-incoherent)—10,125 atoms, 101 frames (1 frame for ensemble average), ∼100 *Q* values.

**Table 3 ijms-25-01547-t003:** Hardware specifications of the used cluster. The CPU was manufactured by Intel Corporation, Santa Clara, USA.

Name	Specification
Number of sockets	2
Number of cores per socket	12
Number of threads per core	2
CPU model	Intel(R) Xeon(R) Silver 4116 CPU @ 2.10 GHz (Skylake-SP)
CPU architecture	x86_64
CPU frequency (minimum)	800.121 MHz
L1d cache (i+d)	32 kB + 32 kB
L2 cache	1024 kB
L3 cache	16,896 kB
Instruction set (vectorization)	SSE2, AVX2 and AVX-512
Memory (per socket)	∼96 GB

## Data Availability

The files used to produce the scattering curves shown in this contribution can be downloaded at https://doi.org/10.5281/zenodo.10037785, We used SLURM queue engines in the cluster for simulations and calculations, which were copied to a local machine for visualization and plotting.

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
