# Peer review of "Computation of X-ray and Neutron Scattering Patterns to Benchmark Atomistic Simulations against Experiments"

_ijms, 2024, doi:10.3390/ijms25031547_

Round 1

Reviewer 1 Report

Comments and Suggestions for Authors

Comments:

This work addresses the crucial need for a program to compute scattering patterns from molecular dynamics simulations, facilitating benchmarking against experimental data. Utilizing the Sassena program, enhancements are introduced to improve computation speed. The study showcases Sassena's versatility across various scattering methods, capturing dynamics from pico- to nanoseconds and characterizing structures spanning from Å to hundreds of nanometers. Notably, the work addresses finite size effects in nanometer-level structures, presenting a method implemented into Sassena to account for this aspect. Overall, the research contributes a valuable tool for accurate comparison between simulations and experimental results, advancing our understanding of material structure and dynamics at the atomic level. The manuscript is well-organized and clearly stated. I would suggest accepting it after the following concerns are addressed. 

Comments on the Quality of English Language

The overall English of this manuscript is good.

Reviewer 2 Report

Comments and Suggestions for Authors

Title: "Computation of X-ray and Neutron Scattering Patterns to Benchmark Atomistic Simulations Against Experiments"

Authors: Arnab Majumdar, Martin Müller, and Sebastian Busch

The article focuses on the enhancement of Sassena for computing scattering patterns from Molecular Dynamics (MD) simulations. It emphasizes the importance of neutron and X-ray scattering experiments in validating atomistic simulations by probing atomic structure and dynamics. The study aims to optimize computation speed, highlighting the need for programs supporting parallelization due to hardware advancements. It discusses various software options, ultimately focusing on enhancing Sassena to address computational limitations in simulating large-scale systems. The article outlines simulations performed on different systems: pure water, a hypothetical spherical solute in solvent, and the protein lysozyme in water. Detailed descriptions of these simulations, the calculation of scattering curves, and methods for probing dynamics are provided. The study exhibits a comprehensive approach to understanding scattering techniques' capabilities in benchmarking simulation accuracy against experimental data.

before considering for publication, I would like the authors to consider the following suggestions:

Clarify the role of Sassena in calculating the incoherent intermediate scattering function for water. Explain Sassena's algorithm briefly or provide a reference for readers to understand it better.

Include more detailed insights into the specific improvements made after employing the Intel MPI compiler, showcasing before and after performance metrics for better comparison.

Perhaps add a brief explanation or reference on SSE2, AVX2, and AVX-512 for readers unfamiliar with these concepts.

Provide a more detailed comparison between the simulated and experimental incoherent scattering functions. Explain the significance of the discrepancies observed in the decay rates for better comprehension.

For clarity, explicitly detail the reasons behind the spurious small-angle scattering signal in calculations, explaining the impact of simulation box size on diffraction patterns.
